# "What are you carrying?" Experiences of mothers with preterm babies in low-resource setting neonatal intensive care unit: a qualitative study

Fatuma Namusoke [1], Musa Sekikubo,[1] Flavia Namiiro,[2] Janet Nakigudde[3]

¹Obstetrics and Gynaecology, Makerere University, Kampala, Uganda
²Paediatrics and Child Health, Mulago National Referral Hospital, Kampala, Uganda
³Psychiatry, Makerere University, Kampala, Uganda

**Correspondence to**
Dr Fatuma Namusoke;
namusokefk@yahoo.co.uk

## ABSTRACT

**Introduction** Babies born preterm often have challenges in feeding, temperature control and breathing difficulty and are prone to infection during the neonatal period. These usually necessitate admission to the neonatal intensive care unit (NICU). Admission to NICU disrupts the mother–baby bonding.

**Objective** This study explored the lived experiences of mothers with preterm babies admitted to NICU in a low-resource setting.

**Study design** This was a qualitative study where 16 participants took part in indepth interviews and 35 in focus group discussions. We included mothers who delivered and were caring for preterm babies at the NICU of Mulago National Referral Hospital.

**Study setting** Data were collected from a public hospital, which works as a district and national referral hospital located in the capital of Uganda.

**Participants** Fifty-one mothers with preterm babies in the NICU were sampled and recruited after informed consent. Data were analysed using manual thematic analysis.

**Results** There were six themes on the experiences of mothers of preterm babies in NICU: constant worry and uncertainty about the survival of their babies, baby feeding challenges, worries of discharge, communication gaps between mothers and nurses, community acceptability and disdain for preterm babies, and financial challenges.

**Conclusions and recommendations** Mothers of preterm babies admitted to NICU in a low-resource setting still need a lot of support other than the medical care given to their babies. Support groups in the hospital and community are recommended to help in dealing with these challenges.

## Strengths and limitations of this study

► The study used qualitative methods with indepth interviews and focus group discussions to document the experiences of mothers with preterm babies.
► To ease sharing of experiences by participants, all researchers involved in data collection were female and one of them was a clinician in the same hospital.
► Further probing where necessary was done to clarify understanding of the experiences.
► Data collection was done in a hospital setting, which may have compromised the quality of data since participants were still receiving care.

In Uganda, preterm delivery accounts for 14% of live births and 28% of neonatal deaths.[5] Despite decreases in under-5 mortality, neonatal mortality rate did not change between 2006 and 2016.[6] More than 50% of all low birthweight babies in Uganda are preterm. In 2011, more than three-quarters of low birthweight babies in Uganda died in the neonatal period.[7]

Additionally, preterm babies have poor body temperature regulation, respiratory distress syndrome and intracranial haemorrhage, which may further compromise their survival.[8] Admission of preterm babies is therefore essential to avert short-term and medium-term complications in this patient category. However, due to limited medical supplies required in the critical care of preterm babies in developing countries, mortality and long-term morbidity are disproportionately high.

Breast milk is the best food recommended for preterm babies[9] and pumping is required as preterm babies have difficulty latching on the breast; it is a costly practice in developing countries.[10] Mothers need support to correctly and hygienically collect breast milk for their preterm babies, but this is hampered by the high patient to nurse ratio.

## INTRODUCTION

Globally, the rate of under-5 mortality decreased significantly from 69.4 to 34.4 per 100 000 live births between 2000 and 2016,[1] but only modestly in developing countries. More than 40% of deaths in under-5 children occur in the neonatal period.[2] Preterm birth is a major cause of morbidity and mortality in the neonatal period[3] and accounts for half of all neonatal deaths.[4] Babies born preterm and survive the neonatal period suffer short-term and long-term complications.

The Mulago National Referral Hospital is the main government maternity hospital serving the metropolitan area of Kampala and receives referrals from district and regional hospitals. In 2019, the population of Kampala metropolitan area was estimated to be 3,138,000 people, 53% of whom are women. The neonatal intensive care unit (NICU) has limited infrastructure and skilled healthcare providers to deal with the high number of high-risk small and sick newborns as reported in healthcare services giving care to the sick/small newborn.[11] Other NICU facilities (mostly level 2) available in Kampala are paid for and include one private not-for-profit hospital and two private hospitals at a limited scale. The admission of a preterm baby to the NICU soon after delivery leads to separation from the mother. This separation presents a dual challenge to the mother coping with her puerperium, nursing a preterm and worrying about the uncertain outcomes of her newborn. It is still not clear what mothers of preterm babies in a low-resource setting are facing given the multiple challenges, which is the subject of this study.

## METHODS

A total of 52 mothers were included and 16 in indepth interviews (IDI) and 4 focus group discussions (FGD) were conducted. The study was conducted in a period of 2 months. The sequence of conducting IDI followed by FGD was to use the FGD to further explore the phenomena identified in the IDI.

### Study setting

The Mulago National Referral Hospital is located in Uganda's capital city and is one of the busiest maternity and neonatal centres in the country. The NICU admits preterm infants and newborns with complications from the hospital as well as referrals from clinics and hospitals around the country. In 2020, the NICU had 4572 admissions, with 51% (2345 of 4572) of preterm babies born within the facility (hospital records). It has a total bed capacity of 49 cots, but usually admits up to three times its capacity, averaging 400 admissions per month, with more than half of them due to prematurity.[12] The team caring for preterm babies includes 6 paediatricians, 19 nurses and 4 nursing assistants. The nurses working at the NICU are trained and dedicated to the unit. The nurses work in a 12-hour shift, with more allocations in the day compared with night duty. The average nurse patient ratio in the NICU is 1:35. Babies admitted to the NICU are assessed for gestational age using the modified Ballard score. During the time of the study, the Mulago National Referral Hospital had its Department of Obstetrics and Gynecology and NICU transferred to Kawempe Division due to the ongoing renovation of the main hospital. The relocation compromised the space available to cater for the same number of neonates. Apart from treatment given, family nurture interventions (FNI) in the NICU are done to prevent the adverse effects of separating mothers from their preterm babies. The FNI offered during the time of study was skin-to-skin care during the time of feeding. Kangaroo mother care services were not offered due to space limitations at Kawempe. The mothers/attendants are sometimes asked to buy drugs and supplies like surfactant, antibiotics and feeding tubes used in the newborn care and cater for their sustenance during the hospital stay.

### Study population and sample

Women who had delivered live babies at less than 37 weeks of gestation and admitted to the NICU were purposively sampled and recruited after informed consent. In depth interviews were conducted on mothers whose preterm baby/babies were still in the NICU at the time of the interview. Mothers whose neonates were very ill or had a congenital abnormality and/or were unable to communicate in the local language (Luganda) or English were excluded from the IDI. Women who were organised into focus groups had their preterm baby/babies in NICU, discharged and brought the baby for review during the study period. All mothers included in the FGD had babies below 3 completed months of age. Mothers who participated in the IDI were excluded from the FGD.

### Study procedure

The IDI and FGD were conducted face to face in a private consultation within hospital premises. Eligible mothers were interviewed between the intervals of feeding the babies. The FGD consisted of 10–12 participants who were identified when they brought their babies for review or treatment by the paediatric team. The patients were approached through the nurse in charge, who acted as the gatekeeper who informed them about the study. All participants approached agreed to participate in the study. However, one woman in an FGD had a neonate that appeared not to be breathing well. This was noted by the obstetrician who was seated in the FGD. This woman was taken out before the group commenced so that her baby could immediately be attended to by the paediatrician on duty. Data collection was done by two investigators (JN and FN) and one note taker with training in qualitative research. FN is an obstetrician gynaecologist in the hospital. All the researchers involved in data collection were female and fluent both in English and Luganda. The interview lasted for 30–40 min. The interview was conducted guided by a semistructured interview guide, which is provided in the online supplemental appendix. All the interviews were audio-taped and transcribed.

The male partners were not present at the time of the interview. Data collection started with collecting information on the demographics of the study participants. Mothers were asked to describe their experiences when they learnt that they would be delivering a preterm baby. They were then asked to reflect on what they thought could have led to delivery of the preterm infant. They were then asked to reflect on the care of their preterm baby at the NICU and the challenges encountered.

In order to further understand the phenomenon, any further questions that arose during the interview were pursued for clarification. Data collection was continued until no new data were generated. Data collection and management were done according to the Consolidated Criteria for Reporting Qualitative Studies.

## Data analysis

Data were analysed using manual thematic analysis. Two of the authors (JN and FN) familiarised themselves with the data from the transcripts after data cleaning and generated preliminary codes. On further familiarisation with the data, patterns in the codes were generated and a further exploration of these data provided us with primary and secondary codes, and then finally themes were generated and defined. Each major theme generated was illustrated using quotations from the women. Data analysis was not sequential but iterative because on numerous occasions the two authors would engross themselves in the data, back and forth until they agreed on the final themes.

## Ethical considerations

Administrative clearance for the study was obtained from the Mulago National Referral Hospital administration. Participants were included after informed written consent. Participants were informed that participation was voluntary, with no direct benefit to the participants or their neonates. The findings of the study could impact on future care of preterm babies and their mothers. Arrangements were made with the hospital counsellor for any of the participants who would become emotionally distressed as a result of the interview. None of the participants required counselling following the interview. In Uganda individuals below the age of majority who are pregnant or have a child are considered as emancipated minors and may independently give consent to participate in research approved by the institutional review committee.[13]

## Patient/public involvement

The participants were not involved in the conception, design or conduct of the study. The public were involved in vetting and final approval of the study protocol. The public have permanent representation on the ethical committees that approved the study. The results of the study will be disseminated to the participants through the media.

## RESULTS
### Demographics

Fifty-one mothers were included both in IDI and FGD, 34 (66.67%) of whom were married and more than 70% of them delivered the baby between 23 and 29 weeks of gestation. Details of other demographic characteristics of the study participants are shown in table 1.

**Table 1** Demographic characteristics of mothers with preterm babies

| N=51 | Frequency | Percentage |
| --- | --- | --- |
| Mother's age in completed years | | |
| ≥18 | 5 | 9.8 |
| 19–25 | 19 | 37.25 |
| 26–35 | 24 | 47 |
| 36–45 | 3 | 6 |
| Marital status | | |
| Married | 34 | 67 |
| Single | 17 | 33 |
| Education in years | | |
| ≤7 | 17 | 34 |
| 8–12 | 20 | 39 |
| ≥13 | 14 | 27 |
| Number of pregnancies | | |
| 1–3 | 33 | 65 |
| 4–8 | 18 | 35 |
| Number of live births | | |
| 1–3 | 41 | 80 |
| 4–8 | 11 | 20 |
| Gestation age at delivery i(weeks) | | |
| 23–29 | 36 | 71 |
| >30 | 15 | 29 |

## Themes

Six themes on women's experiences of having a preterm baby were developed after data analysis. The themes were uncertainty about the survival of a preterm baby, feeding challenges of the baby, worries about care of the baby after discharge, communication gap between mothers and nurses and the perceived insensitivity of nurses, community acceptability and disdain for preterm babies, and financial challenges of having a preterm baby.

### Theme 1: uncertainty about the survival of a preterm baby

A number of women with preterm babies at the national referral hospital were worried and uncertain whether their newly born babies would survive. One mother who had delivered twins illustrated this fear in the following IDI quotation:

> I worry more about taking care of my twins, I do not know if they will live or not though they are under good care by the health workers. (Indepth 12)

One mother describes how the changing weight of her preterm baby was constantly a concern and how she would always run back to the hospital because the baby's weight had decreased:

> I was very worried! I even feared the baby and I was asking myself of how long are we going to be in this type of care or if the child will survive anyway! I was discharged after a week! So you can imagine how

scared I was. But whenever the weight would go down I had to run back to hospital. (FGD 2, respondent 6)

The weight of a preterm baby was constantly a worry to the mothers and they would constantly worry that due to the baby's tiny size the baby would not survive. This is illustrated in the following quotation:

My baby was too tiny that I would even fear to hold it. I would only be told by the other mothers that when you give birth to a premature baby he is taken to the Special care unit and kept there, so I would always worry about my baby's size and ask my self if he will live. (FGD 4, respondent 5)

A mother reported that there was too much death in the neonatal special care unit and that every time her baby was moved to a new crib it was because the baby that had been in that space had died and her baby would be replacing the dead one.

Every time they moved my baby to a different bed, it was replacing one that had passed on and so I kept asking myself, whether my baby would die next? (FGD 2, respondent 1)

Mothers were anxious about taking care of their babies; they had feeding challenges and reported that they could go through all the other challenges if only they were sure that their babies would survive. There was reported agony of not knowing that one's baby was going to survive. This is illustrated in the following quotation:

Going back home is one of our least worries if we only knew that we can access help and assistance. But what is most worrisome is the impending looming death of one's baby! (FGD 4, respondent 5)

### Theme 2: feeding challenges of a preterm baby

Feeding of preterm babies had a number of challenges for the mothers; many of the neonates had to be fed through a tiny feeding tube. These are inserted because most preterm babies have difficulty in feeding. Inserting the feeding tube when a mother is discharged was a challenge. This challenge was coupled with maintaining good hygiene and ensuring the baby does not vomit the milk. This is pointed out in the following quotation by a mother:

Sometimes they discharge children with feeding tubes and that tube requires so much care and good hygiene but also the babies keep vomiting when fed through these tubes. We are not taught well on how to give the milk while at home. (FGD 2, respondent 1)

On the same issue of feeding, another mother with a lot of dismay pointed out:

My baby pulled out the tube and the nurse warned me that it would go into the lungs. However she did not help me put it back! (FGD 2, respondent 5)

Further, some mothers had complaints with keeping the inserted feeding tube in place. This required the mother to express the milk but also to learn to feed the baby using the tubes. This could be a challenge to the mothers, as indicated in the following quotation:

Using a tube to feed the baby is not easy! When the feeding tube comes out you have to run to the hospital and yet the tube easily comes out. (FGD 2, respondent 1)

### Theme 3: worries about care of the baby after discharge

Mothers felt that the clinicians supported them as long as they were still in the hospital. It was very distressing for them to be told that they would be discharged because then they would have to accept the whole burden of taking care of their preterm babies' special needs amidst the insecurities of not knowing what to do in case of any eventuality with the baby. The women felt that they would not have access to immediate help if their preterm babies' well-being deteriorated while at home.

I worry about how I will be with my baby after we have been discharged since I have no idea on how to take care of a premature baby. (Indepth 12)

Two mothers from IDI expressed their lack of knowledge of taking care of preterm babies on their own. This is expressed in the following two quotes:

I am not confident enough when it comes to taking care of premature babies since I have no knowledge. (Indepth 13)

Our babies need a lot of attention and time, yet sometimes you are discharged, you are alone at home and you are even afraid of the baby. (FGD 4, respondent 3)

### Theme 4: communication gap between mothers and nurses and the perceived insensitivity of nurses

There seemed to be a communication gap between the nurses and the mothers. Some mothers complained that when they would want to talk to the nurses regarding the care of their babies, the nurses would ignore them, as illustrated in the following quote:

It is always challenging that the health workers tend to ignore us when we call for their assistance. (Indepth 6)

Poor communication between the clinicians and the mothers was further illustrated on occasions when mothers would be given prescriptions for medicines to buy for their babies. The mothers would assume that since they had purchased the medications, these medications would be administered to the babies. Instances would occur when the mothers would later find out that the medicines that had been requested of them were not administered to babies and no explanations would be given to the mothers. This is pointed out in the following quotation:

I was told to go buy the medication from a pharmacy outside the hospital which I did, but every time I asked the doctor to tell me about the progress of the baby, they wouldn't tell me anything not even about administering the medication that they requested me to buy. (FGD 2, respondent 6)

Further a lack of clear communication was indicated when mothers said that they were lost for what to do for a baby who developed other medical issues. One mother kept on wondering if she should bring back her baby for care because the baby now had developed influenza, as described in the following:

We are given review dates to come back, then my baby developed flu, but it was not yet time to bring the baby back for review! So I kept wondering if I had to come-back earlier or wait for the review date, I had been given. (FGD 4, respondent 2)

### Theme 5: community acceptability and disdain for preterm babies

Mothers with preterm babies are always told to protect their babies from catching infections and they are told not to give their babies to people when they come to visit. When a visitor comes to see the baby and the mother does not give them the baby to see because they have been advised against this, the purpose for which he has come is not fulfilled and this may lead to a confrontation. This stresses the mothers, as shown in the following quotation:

We were told not to always expose our babies because they can easily catch infections, so when I went home, I had to hide my baby. So many people felt offended and they would always complain that I do not want to show them my baby. (FGD 4, respondent 4)

The community where the mothers are coming from appeared to not give as much value to preterm babies as the mothers themselves and this was agonising and insensitive to the mothers. One mother reported that people would ask her what she was carrying even when they well knew that she was carrying a baby:

Back home when most people would see my baby, they would even ask me 'what is that you are carrying?' and yet they very well know you are carrying a baby but they say it just to make you feel bad. But you have no option except to continue tolerating people and putting your baby under the sun to work on the yellowing skin. (FGD 4, respondent 4)

### Theme 6: financial challenges of having a preterm baby

The government hospitals offer free consultation services to mothers and their babies. In cases where medicines and supplies are not available, the patients buy from a private pharmacy. All patients are required to cater for their sustenance during the hospital stay. When a baby is born, there are expected expenditures that many new mothers will prepare for. However, it is hard and expensive to envisage preparing for a preterm baby who has uniquely expensive

needs in terms of time and care and emotionally. Many mothers were worried on how they would meet their financial obligations to the baby and those that they were taking care of before they delivered the preterm baby. This is what they had to say:

I worry about how I will take care of my child financially! I need to have money for diapers, tubes and clothes and yet I have had to put my business to a halt so that I can give time to and take care of my baby! (Indepth 2)

Another mother with other financial responsibilities at home and a need to go back to work said:

I was also worried on how I would now take care of the people at home and the finances to take care of my baby is very hard since my husband is a Taxi driver. Sometimes you need to buy medication, diapers. I also worry that I will not be in position to work immediately since I have to stay home and take care of my baby. (Indepth 12)

What you pay for when you go to the hospital with a preterm baby is way above what a mother would pay for if she had a term baby. Medicines like surfactant used in the care of preterm babies are not on Uganda's essential drug list and are usually costly. Taking care of preterm babies requires prolonged stay in the hospital since the mothers are required to stay in the hospital throughout their admission. This leads to interference in the family's income-generating activities. The financial challenges faced by these mothers are expressed in the following quote:

It has also been a challenge in terms of finances due to the high demands while in hospital because the medications are expensive! (Indepth 8)

## DISCUSSION

This study shows delivery of a preterm baby and the consequent admission to the NICU in a low-resource setting is an emotionally packed period with various challenges. The experience of mothers during admission in the NICU shaped the way they looked at their surviving baby. These results are important in highlighting the unique challenges faced by mothers of preterm infants in NICU in a low-resource setting.

Mothers were worried about the survival of their babies. Having their preterm baby in NICU and looking at how other babies were dying left mothers in suspense about the survival of their preterm infants. This is consistent with previous studies, which indicated that mothers of hospitalised infants are worried about their survival.[14 15] Looking at the changes in the weight of the baby with time compounded their anxiety as most of the preterm neonates lose weight before gaining. The risk of death of preterm babies in low and middle income countries continues into the postneonatal period.[16] Survival of preterm neonates is less in developing countries, where facilities for prevention of mortality in the antenatal and early neonatal period are largely lacking.

Our findings indicate that mothers were generally challenged about feeding their preterm babies. Preterm babies have an immature digestive system, which leads to many feeding problems. Preterm babies frequently cannot coordinate suckling,[17] swallowing and breathing, and tend to have gastric residues due to incomplete emptying, gastro-oesophageal reflux and abdominal distension.[18] In order to overcome these problems, preterm babies receive gavage feeding, where a feeding tube is placed through the nose or mouth to the stomach. This allows for easy monitoring of feeding of the preterm infant. The use of nasal gastric feeding tubes has been associated with higher rates of central apnoea than oroenteric feeding.[19] The oroenteric feeding, however, has the challenge of securing the feeding tube in place. Preterm babies are given feeding tubes, which are perceived by mothers to hurt their vulnerable infant.

Mothers were constantly worried about discharge of the baby. They doubted their ability to look after their preterm without the supervision of a health worker. Previous studies have shown that parents of preterm babies get a lot of anxiety and worry when it comes to discharge of the baby.[20 21] There are no standardised criteria for discharge of preterm babies and developing this has been suggested as one of the ways to decrease anxiety of mothers.[22] Giving clear guidelines on criteria for discharge to mothers may reduce the anxiety associated with it.

Some mothers of preterm babies felt that the nurses caring for their babies were insensitive about the challenges they were facing while taking care of their babies. A study done in Ghana showed that where mothers are not able to get the support from health workers in care of their preterm they resorted to peer support.[15] Babies in the NICU are cared for, but mothers may not have good emotional support and information, which would affect their perception of care.[23] Crowded NICUs are constrained by the time a single nurse/clinician will spend with every baby and mother, which leads to prioritising the very ill. To overcome this problem group sessions regarding care of newborns are given and not for individual mothers. Using this approach, there may not be an opportunity to assess individual needs and gaps for each mother.

Participants in this study felt that the health workers were not communicating well with them nor giving them time to ask any questions regarding the condition of the baby and the treatment given. Previous studies done in developed countries have alluded to obtaining similar research findings, where women caring for preterm babies in the NICU valued being listened to and having a formal communication with health professionals.[24 25] The communication problem in this setting may be due to the high patient load with less than optimal staffing levels in the public health setting.[26] Increased workload of nurses in the NICU has been associated with missing nursing care previously.[27] Giving the right information to mothers with preterm labour by health workers helps to empower them in decision-making.[28] Lack of communication and indifference of health workers leave mothers with a preterm baby helpless since they are in charge of their babies.

The community where the mothers come from appeared not to give as much value to preterm babies or the challenges the mothers were going through. Mothers felt that the community and family members did not give them support when they delivered a preterm baby. Our findings were unique when it came to community acceptability of a preterm baby. There is no study where mothers of a preterm baby were worried about what the other members of the community thought about their baby. Uganda is one of the countries in Africa with high fertility rate and high regard for the number of children.[29] Delivering a child is taken as an event in the community, who feel that they have a right to see the baby all the time. Most studies looking at the experiences of mothers with preterm babies are done in developed countries, where close family members are involved in childcare. Where all concerned have to see and care for the baby, a preterm baby ceases to contribute to the oneness of the community because the baby will be kept away for health reasons and people may not even see the baby until it is safe to have him/her out.

Mothers were faced with financial challenges as they looked after their preterm babies. Delivering a preterm infant has been found to present a high economic burden on the family through direct medical and non-medical costs as well as lost working days.[30] Mothers are faced with many financial obligations and have to invest a lot of time while taking care of their preterm infant, as has been described.[15] These challenges are worse in financially constrained population in a low-resource setting.

## Limitations

The results of this study may not be generalised to all mothers of preterm babies in low-resource settings. At the time of data collection, the facility had been shifted to another unit to allow for renovations in the hospital. The medical team and equipment used in the care of the newborns were not affected by the shifting. This study, however, presents the experiences of mothers in resource-limited settings seeking care in a public healthcare facility. Mothers of preterm babies were still receiving care in the facility, which could have biased the results.

## Conclusions

The journey of a woman with a preterm baby was anxiety-provoking due to uncertainties in survival, challenges in care, communication problems with health workers and the harsh treatment of the community following discharge.

## Recommendation

We recommend that mothers attending the antenatal clinic are educated about preterm birth and how this can be handled. We recommend that issues regarding financial preparation, including saving for the expenses that come at the time of delivery, should be included in antenatal health talks. Financial empowerment of women of reproductive age and educating female children in these settings are highly recommended strategies to avert the challenges. Mothers attending to babies in the NICU could be organised in

support groups as they go through the stressful time. Family aturenurture interventions aiming at supporting mothers and their preterm infants are highly recommended as a solution to some of the challenges. A pilot study in a low-resource setting is recommended before rolling it out owing to the diversity of challenges faced by mothers of preterm babies. We recommend increased funding for the health sector in an effort to decrease the patient to nurse/doctor ratio.

**Acknowledgements** We would like to acknowledge the research participants who provided information on which this writing is based. We are very grateful to the partners and family that supported the mothers during the hospital stay. We would also like to acknowledge Professor Derek Pomeroy for proof-reading the article.

**Contributors** All authors contributed to conception of the study. FNamu and JN collected and analysed the data. FNamu, MS, FNami and JN have critically reviewed the manuscript for intellectual content. All authors approved the final version of the manuscript and are accountable for all aspects of the work presented here.

**Funding** The Bill and Melinda Gates Foundation supported this work through the 'Preterm Birth initiative' project to the University of California San Francisco (grant number OPP1107312).

**Competing interests** None declared.

**Patient consent for publication** Not required.

**Ethics approval** Prior to initiating data collection, the study was approved by the Makerere University School of Medicine Research and Ethics Committee (REF 2018-092) and the Uganda National Council for Science and Technology (HS 2457). All study participants provided informed consent and those below the age of majority were considered as emancipated minors.

**Provenance and peer review** Not commissioned; externally peer reviewed.

**Data availability statement** Data are available upon reasonable request.

**ORCID iD**
Fatuma Namusoke http://orcid.org/0000-0002-3610-7817

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
