## [Reviewer comments · BMJ Open]

ARTICLE DETAILS

TITLE (PROVISIONAL)	'What are you carrying?': `Experiences of mothers with preterm babies in low resource setting neonatal intensive care unit - a qualitative study
AUTHORS	Namusoke, Fatuma; Sekikubo, Musa; Namiiro, Flavia; Nakigudde, Janet

VERSION 1 – REVIEW

REVIEWER	Welch, Martha G. Columbia Univ Coll Phys
REVIEW RETURNED	23-Oct-2020

GENERAL COMMENTS	In their manuscript, "What are you carrying?": `Lived experiences of mothers with preterm babies in low resource setting neonatal intensive care unit - a qualitative study, Namusoke and coauthors have used focus groups and interviews to assess mothers of preterm infants. This is an important paper in that it highlights the challenges of premature birth in a population that has no voice. This paper is suitable for publication in BMJ Open with the following suggestions for revision:  • "Lived experiences is redundant. Authors should consider dropping the word "lived". • First 2 article summary bullets are repetitive. Drop second bullet. • Page 3, line 47. The word "purposively" is not needed. • While the article is generally well written, there are sections of the methods, results and discussion that are highly repetitive and can be reduced for simplicity • Can the authors include whether any nurture interventions are available in the NICU or generally in the hospital setting (e.g., kangaroo care, skin-to-skin care). • Please include in the discussion interventions or services that might be brought to this hospital setting. For example, in the Family Nurture Intervention, a program that supports mother-infant emotional connection, family support and overall consideration of the premature infants relational needs, may provide a low cost structure to improve premature infant outcomes and maternal support (e.g., [1]) [1] Welch MG, Firestein MR, Austin J, Hane AA, Stark RI, Hofer MA, et al. Family Nurture Intervention in the Neonatal Intensive Care Unit improves social-relatedness, attention, and neurodevelopment of preterm infants at 18 months in a randomized controlled trial. J Child Psychol Psychiatry. 2015;56:1202-11.
---

	The reviewer provided a marked copy with additional comments. Please contact the publisher for full details.
--	--

REVIEWER	Edwards, Erika Vermont Oxford Network
REVIEW RETURNED	14-Nov-2020

GENERAL COMMENTS	Generally, this manuscript is interesting and the findings are important. I find the introduction the most challenging. Assume that your readers do not know anything about NICU care in Uganda. Set the stage for them. It will help us understand why did you did this research. It may also help us understand what this research adds to the literature. Abstract  Line 32: The language could be more precise. Babies born preterm are too immature to survive without admission to a neonatal intensive care unit. They do not suffer complications as a result of being premature; in fact, some would argue that they suffer complications from being admitted to a NICU. Instead, they face difficult challenges as a result of being premature. Line 50: I count five areas of concern unless the first one is actually two? Introduction  I advise using local data when possible to put the research in context. Are mortality, preterm birth rates, preterm mortality rates, and NICU admission rates available for Uganda? How many preterm infants are born and survive to discharge every year? Additionally, it would be interesting for readers to know something about NICUs in Uganda; how many in total, how many are public, etc. For this manuscript in particular, it would be good to know something about nursing at this hospital. What is the average NICU nurse-to-patient ratio? Do nurses rotate to different units or are they dedicated to the NICU? Line 94: How is preterm defined here? <37 weeks GA? Line 96: It is not culturally acceptable in Uganda? (It is culturally acceptable in the U.S.) Methods  Line 115: There is a word missing – preterm infants. Line 169: “audio taped” comes before “transcribed” Line 181: What is a “modified grounded theoretical basis”? The “Study procedure” sub-section is repetitive. Discussion  The first paragraph of the discussion should reiterate your findings. Here is a good place to say: “We identified six themes...”. You do not need to cite other people’s work in this first paragraph. Show off what you learned! What did you learn that was new, or surprising? The sentence at 402-404 needs a citation. (Or, we need a better understanding of the preterm mortality rate in Uganda.) In regard to the sentence at lines 434-435: Was this study (ref. 18) done in Uganda? Please specify. Line 466: What did you learn about the financial challenges? Recommendation  Are there any recommendations about or for the nursing and medical staff?
---

REVIEWER	Clarkson, Gina Idaho State University, School of Nursing
REVIEW RETURNED	18-Nov-2020

GENERAL COMMENTS	Thank you for giving me the opportunity to review this manuscript. I hope the following suggestions are helpful. Abstract- What do you mean by 'desired' separation? Total number of participants should be mentioned here with numbers of in-depth interviews and focus groups with each group number of participants. Introduction- I would like to see information in the introduction more specific to the circumstances and context of the women who participated in this study. Some information is here about breastfeeding reluctance, but please tell us more about the community and the NICU in a low resource setting. Where is the literature review? Methods- How many phenomenological interviews vs. focus groups? What was the timeframe of the interviews? How were the women recruited? You use past tense...when were the babies in the NICU and when did you interview the women? Focus group women were interviewed while they were bringing the babies back, I see. Is that at 2 months of age in this country? You held group interviews after the in-depth interviews...why? Setting- You have information about the transfer of the NICU here, which is fine, but some of what you have written here belongs in the discussion as a limitation in recruitment. By 'patients cater to their sustenance' do you mean the babies too? Subjects- Were the same women who had been interviewed also included in focus groups? Study procedures- You use the word 'parents' but stated just above that the fathers were not present. I think you should consistently use the word 'mother/s' instead since fathers were not a part of this study. Did the focus groups provide new information? When you say all mothers with preterm babies from 1 day to 6 months, do you mean the age of the child or the amount of prematurity? Please clarify. Data analysis- You do not need to list the themes here, they should be in the results. Ethical considerations- You have subject recruitment procedures here. Please move this information to the subjects section. Please change 'would get emotional distress' to 'would become emotionally distressed'. Please speak to the ethics of involving persons under the age of 18 in your country. Table 1- Can the layout of this table be condensed more? How many participants were under age 18? Please differentiate headings and subheadings better. Themes- please clarify that these themes are related to women having a preterm baby in a low resource setting. Theme 1- This theme could be a manuscript in and of itself. I cannot imagine the horror and anxiety that these mothers must be going through. How much death is there specifically in your country and this NICU? Some of this information should be in the introduction.
---

	Theme 2- I would like to have some information about when babies are being discharged and the discharge criteria in the setting section. Discussion I would place the information about mothers in Sweden elsewhere. Uncertainty about survival- It seems you are using studies with participants from areas with more resources for preterm infants. Are there any studies with mothers of preterm infants in low resource settings with which you can compare your findings? Feeding challenges- You aren't really comparing your results to other studies here, it seems like you are including information that should have been in your introduction or literature review. What do other studies report the mothers are feeling and saying and doing about feeding challenges in a low resource setting? Worries about discharge- Please change the word 'get'. "The mothers knowing criteria for discharge"... Please explain this sentence more and cite relevant literature. Perceived insensitivity- You put excellent recommendations for practice here, please move this information to the appropriate section. Please review relevant literature and provide a comparison of your findings here. Communication gap- The last two sentences in this section are more relevant to conclusions or practice recommendations sections. Community- The information here is very good but should be moved to the introduction, I believe. This section should compare relevant literature. If no comparing literature, is this the first time this information is being reported? Limitations- You say that data is being collected while the baby is in the NICU, but reported elsewhere that the focus groups were conducted after discharge. Conclusions- What was new or innovative about this research study? Where should future research go? Recommendations- You had great recommendations in a few of your discussion sections for the different themes. While all of your themes are important, some are obviously more detrimental to the psychological welfare of the mother and her relationship with her baby. What are your concrete recommendations for 'uncertainty about survival' and 'community acceptability'? Are support groups enough? Is it possible to work on community support and changing attitudes of the community? Could more funding be brought in to the NICU? It seems these two themes alone could be a start to two different programs of research. I think you might be better off splitting this manuscript into during the hospital stay and after the hospital stay. The context for each group of women is so different. There is so much important information to report and I think it's a disservice to try to report all of this in one manuscript.
--	--

VERSION 1 – AUTHOR RESPONSE

Reviewer #1's comment	How it was addressed	Where it appears in the text
General Comment:		
Lived experiences is redundant. Authors should consider dropping the word "lived".	Lived has been dropped throughout the manuscript	N/A
First 2 article summary bullets are repetitive. Drop second bullet.	The repeated bullet has been deleted	N/A
Page 3, line 47. The word "purposively" is not needed	It has been deleted	N/A
While the article is generally well written, there are sections of the methods, results and discussion that are highly repetitive and can be reduced for simplicity	Critical revision has been done to eliminate the repetitiveness	N/A
Can the authors include whether any nurture interventions are available Please include in the discussion interventions or services that might be brought to this hospital setting. For example, in the Family Nurture Intervention, a program that supports mother-infant emotional connection, family support and overall consideration, of the premature infants relational needs, may provide a low cost structure to improve premature infant outcomes and maternal support	At the time of data collection skin to skin contact was practiced-included in study setting Included as a recommendation from the study	Line 152-156 Line 522-524
Reviewer 2		
The language could be more precise. Babies born preterm are too immature to survive without admission to a neonatal intensive care unit. They do not suffer complications as a result of being premature; in fact, some would argue that they suffer complications from being admitted to a NICU. Instead, they face difficult challenges as a result of being premature	Revised as suggested	Line 31-36

Line 50: I count five areas of concern unless the first one is actually two? Introduction	There are six themes mentioned	Line 54-57
I advise using local data when possible to put the research in context. Are mortality, preterm birth rates, preterm mortality rates, and NICU admission rates available for Uganda? How many preterm infants are born and survive to discharge every year?	Included in the introduction and study setting	Line 81-86 Line 131-136
Additionally, it would be interesting for readers to know something about NICUs in Uganda; how many in total, how many are public, etc.	Included in the introduction section	Line 97-108
For this manuscript in particular, it would be good to know something about nursing at this hospital. What is the average NICU nurse-to-patient ratio? Do nurses rotate to different units or are they dedicated to the NICU?	Nursing in the hospitals and the ratios are mentioned in introduction. The shifts are included in the study setting.	Line 130-133
Results:		
Line 94: How is preterm defined here? <37 weeks GA?	Delivery before 37 completed weeks of gestation confirmed using Ballard score.	Line 146-147
It is not culturally acceptable in Uganda? (It is culturally acceptable in the U.S.)	This section has been revised to reflect the costs of pumping breast milk.	Line 94-95
Methods Line 115: There is a word missing – preterm infants. • Line 169: “audio taped” comes before “transcribed” • Line 181: What is a “modified grounded theoretical basis”? The “Study procedure” sub-section is repetitive	Added Changed It has been revised to read modified grounded theory Critical revision done to eliminate the repetition	Line 197 Line 173
Discussion:		

The first paragraph of the discussion should reiterate your findings. Here is a good place to say: “We identified six themes...”. You do not need to cite other people’s work in this first paragraph. Show off what you learned! What did you learn that was new, or surprising?	This has been revised as suggested	Line 398-405
The sentence at 402-404 needs a citation. (Or, we need a better understanding of the preterm mortality rate in Uganda.)	Included the citations	line 424-435
In regard to the sentence at lines 434-435: Was this study (ref. 18) done in Uganda? Please specify	The study was done in England. It has been specified in the manuscript	Line 458
Line 466: What did you learn about the financial challenges?	Included in the recommendations	Line 499-503
Recommendation [ ] Are there any recommendations about or for the nursing and medical staff? .	Included in the recommendations increase funding in the health sector to bridge the gap	Line 512-514
Reviewer 3		
Abstract- What do you mean by ‘desired’ separation? Total number of participants should be mentioned here with numbers of in-depth interviews and focus groups with each group number of participants	Has been revised to remove ‘desired’ Numbers included have been indicated in the abstract	Abstract; 31-40 Line 41-43

Reviewer #3's comment	How it was addressed	Where it appears in the text
General Comment:		
Introduction- I would like to see information in the introduction more specific to the circumstances and context of the women who participated in this study. Some information is here about breastfeeding reluctance, but please tell us more about the community and the NICU in a low resource setting. Where is the literature review?	Introduction has been revised critically to cater for the recommendation.	Line 92-96 Line 97-107
Methods- How many phenomenological interviews vs. focus groups? What was the timeframe of the interviews? How were the women recruited? You use past tense...when were the babies in the NICU and when did you interview the women? Focus group women were interviewed while they were bringing the babies back, I see. Is that at 2 months of age in this country? You held group interviews after the in-depth interviews...why?	Sixteen in depth interviews and four focus group discussions were conducted Data was collected in a period of 2 months : 53 participants Recruited in sequence, in-depth interviews then FDGs . This allowed further explore the phenomena from IDI	Line 112 Line 113-114 Line 115-116
Setting- You have information about the transfer of the NICU here, which is fine, but some of what you have written here belongs in the discussion as a limitation in recruitment. By 'patients cater to their sustenance' do you mean the babies too?	This has been included in the limitations of the study' Parents or guardians provide their sustenance	Line 467-468 Line 143

Subjects- Were the same women who had been interviewed also included in focus groups? Study procedures- You use the word 'parents' but stated just above that the fathers were not present. I think you should consistently use the word 'mother/s' instead since fathers were not a part of this study. Did the focus groups provide new information? When you say all mothers with preterm babies from 1 day to 6 months, do you mean the age of the child or the amount of prematurity? Please clarify.	No Revised Yes Yes , age of the child	Line 153-155 Line 157 Presented in the results section LINE 119-120
Data analysis- You do not need to list the themes here, they should be in the results.	This has been revised mentioned I results section only.	N/A
Ethical considerations- You have subject recruitment procedures here. Please move this information to the subjects section. Please change 'would get emotional distress' to 'would become emotionally distressed'. Please speak to the ethics of involving persons under the age of 18 in your country.	These are emancipated minors according to Ugandan law	Line 209-210
Table 1- Can the layout of this table be condensed more? How many participants were under age 18? Please differentiate headings and subheadings better.	Included in Table 1	Line 220

VERSION 2 – REVIEW

REVIEWER	Edwards, Erika Vermont Oxford Network
REVIEW RETURNED	26-Apr-2021
GENERAL COMMENTS	Thank you for this excellent revision. I particularly like the Recommendations. No additional changes.